# Pharmacological Properties of *Jaeumgeonbi-Tang* on Redox System and Stress-Related Hormones in Chronic Subjective Dizziness: A Randomized, Double-Blind, Parallel-Group, Placebo-Controlled Trial

**DOI:** 10.3390/ph15111375

**Published:** 2022-11-09

**Authors:** Chan-young Kim, Hyeong-Geug Kim, Hye Won Lee, In-Chan Seol, Yoon-Sik Kim, Hyung Il Choi, Miso S. Park, Ho-Ryong Yoo

**Affiliations:** 1Department of Cardiology and Neurology of Korean Medicine, College of Korean Medicine, Daejeon University, Daejeon 34520, Korea; 2Department of Biochemistry and Molecular Biology, Indiana University School of Medicine, Indianapolis, IN 46202, USA; 3KM Convergence Research Division, Korea Institute of Oriental Medicine, Daejeon 34504, Korea; 4Department of Alternative Medicine, Graduate School, Kyonggi University, Seongnam 13557, Korea; 5Clinical Trial Center, Daejeon Korean Medicine Hospital of Daejeon University, Daejeon 34520, Korea

**Keywords:** *Jaeumgeonbi-Tang*, chronic subjective dizziness, clinical trial, redox system, stress hormone, medicinal herbal plants

## Abstract

*Jaeumgeonbi-Tang* (JGT), a traditional herbal medicine, has been used to treat dizziness and vertigo in Korea and China for hundreds of years. The purpose of this study was to evaluate the pharmacological properties of JGT in chronic subjective dizziness (CSD) patients. A randomized, double-blind, parallel-group and placebo-controlled trial was performed with a total of 50 CSD patients. The patients were randomly assigned to one of two groups: JGT or placebo (*n* = 25 for each). All participants received the treatment (placebo or JGT, 24 g/day) for 4 weeks. We analyzed the serum levels of oxidative stressors, antioxidants, and stress hormones. Serum levels of lipid peroxidation, but not nitric oxide, were significantly decreased in the JGT group. JGT not only prevented the decline of serum total glutathione contents and total antioxidant capacity, but it also increased superoxide dismutase and catalase activities. Serum levels of stress hormones including cortisol, adrenaline, and serotonin were notably normalized by JGT treatment, but noradrenaline levels were not affected. Regarding the safety and tolerability of JGT, we found no allergic, adverse, or side effects in any of the participants. JGT showed beneficial effects on CSD patients by improving redox status and balancing psycho-emotional stress hormones.

## 1. Introduction

Dizziness is one of the most common ambiguous, subjective symptoms, and its severity is difficult to quantify. Many dizzy patients find it difficult to express their symptoms and are frequently misdiagnosed [1]. The classification of dizziness quality (e.g., vertigo, presyncope, disequilibrium, or light-headedness), which has been previously focused on by clinicians, has a lack of clinical usefulness. As a result, recent transdisciplinary studies have focused on differentiating dizziness based on its duration (e.g., acute, episodic, or chronic) or pathophysiological mechanism [2,3]. Over the last three decades, many researchers have classified dizziness with specific shared core physical features as chronic functional disorders, which include phobic postural vertigo (PPV) [4], space-motion discomfort (SMD) [5], visual vertigo (VV) [6], and chronic subjective dizziness (CSD) [7]. Efforts to characterize the dizziness served as the foundation for defining persistent postural perceptual dizziness (PPPD), which was the result of a consensus reached by the Bárány Society in 2014 [8].

CSD, which further clarified the concepts of phobic postural vertigo, is characterized by continual non-vertiginous dizziness or unsteadiness that lasts for months or years. Vestibular, visual, and proprioceptive motion stimuli may aggravate symptoms, and patients may find it difficult to tolerate complex visual environments (e.g., shopping malls) or precision visual tasks (e.g., reading) [9,10]. CSD is usually triggered by episodes of vertigo or unsteadiness of vestibular, neurological, or psychiatric origin [11]. Patients with CSD, on the other hand, have no active vestibular deficits on neuro-otological or laboratory examination. There are currently only a few effective treatments for CSD. Selective serotonin reuptake inhibitors (SSRIs) and serotonin noradrenaline reuptake inhibitors (SNRIs) have been recommended to treat CSD. However, one out of every five CSD patients discontinues SSRIs due to sensitivity to their side effects [11,12]. Therefore, more research into effective treatments for CSD is needed.

According to *Donguibogam*, a traditional Korean Medicine (TKM) literature, *Jaeumgeonbi-Tang* (JGT)—herbal medicine, also known as *Ziyinjianpi-Tang* in Chinese—has been popularly prescribed for hundreds of years to treat dizziness, vertigo, anxiety, depression, and emotional stresses. Previous studies have revealed some of the pharmacological effects of JGT in preclinical animal models and clinical trials [13,14,15,16]. However, the possible underlying mechanisms of JGT are still unclear and no attempts have yet been made to observe the biomarkers of human peripheral blood. A recent study demonstrated well the pathological features of CSD, which are intimately linked to the endogenous redox (reduction–oxidation) system [17]. Therefore, we sought to investigate the pharmacological effect and underlying mechanisms of JGT on patients with CSD. The study mainly focused on the serum levels of oxidative stress indices and stress hormones. Furthermore, the hematological and biochemical parameters of the whole blood samples of the patients were analyzed to determine the safety and tolerability of JGT.

## 2. Results

### 2.1. JGT Treatment Reduced Serum TBARS Levels, but Not NO Levels in CSD Patients

First, we investigated the effect of JGT on oxidative stress biomarkers. We measured the serum levels of nitric oxide (NO), which acts as a free radical and can directly or indirectly cause cell damage. The baseline serum NO levels in the JGT group were 10.54 ± 12.22 μM and in the placebo group were 9.24 ± 8.08 mM. These levels of NO in the serum were decreased after 4 weeks of JGT or placebo administration in both groups, but not significantly (8.43 ± 6.58 μM for the JGT group, and 7.66 ± 3.23 μM for the placebo group, respectively, Figure 1A). Next, the serum levels of malondialdehyde (MDA, an end product of oxidation, measured by thiobarbituric acid reactive substances (TBARS) assay), were measured. Before administration, the JGT group had serum MDA levels of 10.89 ± 1.24 μM, while the placebo group had levels of 10.22 ± 2.82 μM. After 4 weeks of the JGT or placebo administration, serum MDA levels, measured by TBARS assay, were significantly decreased in the JGT group (6.04 ± 0.95 μM, *p* = 0.005 for before vs. after in the JGT group, Figure 1B) as compared with the placebo group (8.30 ± 1.54 μM; *p* = 0.027 for placebo vs. JGT, Figure 1B).

### 2.2. JGT Treatment Increased Serum Endogenous Antioxidant Components in CSD Patients

To verify the antioxidant effects of JGT on CSD patients, we looked at serum markers of endogenous antioxidant components such as total glutathione (GSH), Trolox equivalent antioxidant capacity (TEAC), and catalase and superoxide dismutase (SOD) activities. GSH is a small peptide and a powerful endogenous antioxidant; it can easily quench protein adducts in oxidative stress. GSH can restore physiological homeostasis and normalize abnormal redox status. Figure 1C shows that after 4 weeks of JGT administration, the total GSH content in serum levels increased significantly (from 51.58 ± 46.62 to 60.93 ± 65.73 μM for the JGT group and from 63.28 ± 17.18 to 48.33 ± 23.44 μM for the placebo group, respectively; *p* = 0.022 for placebo vs. JGT). Furthermore, after 4 weeks, serum TEAC levels in the JGT group were significantly improved compared to the placebo group (from 220.92 ± 17.69 to 233.41 ± 13.74 μM for the JGT group (*p* = 0.002 for before vs. after in the JGT group, Figure 1D), and from 229.64 ± 9.76 to 227.78 ± 1.51 μM for the placebo group, respectively; *p* = 0.023 for placebo vs. JGT, Figure 1D).

The enzymatic antioxidant components, catalase, and superoxide dismutase (SOD) are the most pivotal elements in maintaining physical redox balance. Under pathological conditions, increasing the activities of these two enzymes can protect biological organisms from free radical-induced cellular damage. After 4 weeks of JGT or placebo administration, serum catalase activities increased from 77.08 ± 12.53 to 89.79 ± 26.40 units/mL in the JGT group (*p* = 0.034 for before vs. after in the JGT group, Figure 1E) while the serum catalase activities decreased from 75.54 ± 22.04 to 64.62 ± 18.65 units/mL in the placebo group (*p* = 0.034 for placebo vs. JGT, Figure 1E). Serum SOD activities, meanwhile, increased from 8.95 ± 5.10 to 15.9 ± 2.15 units/mL in the JGT group (*p* = 0.018 for before vs. after in the JGT group, Figure 1F), and from 10.9 ± 7.25 to 11.55 ± 8.95 units/mL in the placebo group (*p* = 0.047, Figure 1F).

### 2.3. JGT Treatment Reduced Serum Cortisol and Adrenaline While Increasing Serotonin in CSD Patients

We measured serum levels of cortisol, which is a general stress hormone secreted from the adrenal cortex as a result of the hypothalamus–pituitary–adrenal (HPA) axis activation. In this study, serum cortisol levels in the JGT group significantly decreased after 4 weeks of JGT administration (from 236.35 ± 32.74 to 202.31 ± 25.65 ng/mL, *p* = 0.001 for before vs. after in the JGT group), compared to the placebo group (from 225.01 ± 28.27 to 244.68 ± 30.12 ng/mL, *p* = 0.001 for placebo vs. JGT, Figure 2A).

Adrenaline and noradrenaline are well-known stress response hormones that are released in response to mental or physical stress via abnormal sympathetic nerve activation. We found that the JGT administration significantly ameliorated serum adrenaline levels in the JGT group (from 621.21 ± 72.36 to 367.99 ± 87.99 ng/mL, *p* = 0.001 for before vs. after in the JGT group, Figure 2B) compared to the placebo group (from 606 ± 197.92 to 473.12 ± 87.75 ng/mL, *p* = 0.002 for placebo vs. JGT, Figure 2B). However, the JGT administration did not significantly reduce serum noradrenaline levels (Figure 2C).

Serotonin is the most crucial neurotransmitter in the regulation of emotional stress and neurodegenerative diseases. Serum levels of serotonin improved after 4 weeks of the JGT administrations in the JGT group (from 91.58 ± 242.50 to 135.54 ± 120.35 nM/mL, *p* = 0.087 for before vs. after in the JGT group, Figure 2D) compared to the placebo group (from 119.99 ± 267.29 to 80.31 ± 99.37 nM/mL, *p* = 0.047 for placebo vs. JGT, Figure 2D).

### 2.4. Safety Assessment of JGT

In terms of medication safety, we obtained safety test results in the serum biochemistry and peripheral blood levels. First, we examined the potential hepatotoxicity of JGT administration during the experimental period. Serum levels of liver enzymes (such as aspartate aminotransferase (AST), alanine aminotransferase (ALT), alkaline phosphatase (ALP), and gamma-glutamyl transpeptidase (γ-GTP)), total protein, albumin, total cholesterol, triglyceride, glucose, creatinine, and blood urea nitrogen (BUN) were not altered from the baseline to the endpoint, in both groups (Appendix A). In addition, there were no abnormalities in red blood cells (RBC), white blood cells (WBC), neutrophils (segment), lymphocytes, monocytes, hemoglobin, hematocrit, platelets, erythrocyte sedimentation rate (ESR), mean corpuscular volume (MCV), mean corpuscular hemoglobin (MCH), red cell distribution width (RDW) (Appendix A).

We further checked the effect of JGT on liver and kidney parameters in reference to the ages of the subjects. Because the subjects’ ages ranged from 20 to 62, we divided the subjects into two groups: those aged 20 to 45 and those aged 46 to 62. We used Wilcoxon signed rank test to determine whether liver and kidney parameters in each group differed before and after taking JGT. After 4 weeks of intervention in the JGT group, the increase in total protein in the blood was statistically significant in the 20–45 group, but the change was within the normal range. After 4 weeks of intervention in the placebo group, the increase in total protein in the blood and the decrease in A/G ratio were statistically significant in the 46–62 group, but they were also within the normal range. Other items revealed no statistically significant results. The above differences may have been statistically significant because of the results of multiple tests.

## 3. Discussion

Clinically, CSD is difficult to diagnose, and many CSD patients are treated empirically for their symptoms. The first necessary step is to investigate the medications that the patients are taking and try to discontinue them if there is any medication that causes dizziness. Antihistamines such as meclizine and dimenhydrinate, which have an anticholinergic effect, should be tried first because they can effectively control dizziness caused by peripheral vestibular problems. Furthermore, benzodiazepines such as diazepam, clonazepam, or lorazepam can be used to treat central vestibular disorder symptoms by increasing the inhibitory effect of GABA (gamma-aminobutyric acid), a representative inhibitory neurotransmitter. However, these medications can have several side effects. For patients complaining of uncontrolled dizziness, these drugs are frequently administered for an extended period, while they should only be used in acute situations, as they inhibit vestibular compensating mechanisms, slowing the patient’s functional recovery [18,19].

Antidepressants such as selective serotonin reuptake inhibitors (SSRI) and serotonin–noradrenaline reuptake inhibitors (SNRI) were previously used to treat dizziness associated with anxiety. However, these drugs are being recommended for the treatment of chronic functional dizziness, such as PPPD or CSD, regardless of anxiety. Nevertheless, because these patients, like those with other functional disorders, are more susceptible to the side effects of these medications, dosing must begin slowly and usually at less than half of the recommended dose ranges for depression [11,12].

Chronic dizziness has a negative impact on the quality of life, particularly in the elderly. While there are currently unmet needs for an effective and safe treatment for chronic dizziness [20], JGT has been clinically effective among herbal medicines and has been used for hundreds of years in Korea to treat various types of dizziness. As a result, we carried out this clinical trial to confirm how JGT works in CSD patients. To obtain scientific evidence of the efficacy and pharmacological properties of JGT on CSD, we conducted a randomized, double-blind, parallel-group, placebo-controlled trial. In the current study, we followed the well-designed clinical trial protocol of a previous TKM-based study [21], which yielded positive results. Additionally, previous studies have suggested that oxidative stress may play a role in the progression of CSD [22,23]. On that account, we concentrated on oxidative stress biomarkers and endogenous antioxidant components to validate the pharmacological properties of JGT.

JGT administration for 4 weeks (8T TID, a total of 24 g a day) significantly ameliorated serum MDA levels when compared to the placebo administration (Figure 1B, *p* = 0.027 for changed values compared to the placebo group and *p* = 0.005 for before vs. after in the JGT group, respectively), but did not affect serum NO levels (Figure 1A). Oxidative stress directly or indirectly mediates the pathogenesis of a wide range of illnesses by damaging DNA, cells, or tissue [24,25]. Excessive NO generation provokes cellular or tissue oxidation via lipid peroxidation, which has been linked to a variety of diseases such as neurodegenerative disorders (e.g., Alzheimer’s disease and Parkinson’s disease), type 2 diabetes, and cancer [25,26,27,28]. In this study, even though the JGT administration did not affect serum NO levels, it did have an ameliorating effect on oxidative stress.

Most biological organisms have a well-equipped defense system called the antioxidant system, which includes non-enzymatic and enzymatic antioxidant components to maintain homeostasis in the redox system [29]. The representative non-enzymatic component is GSH, a peptide-derived potent antioxidant component. GSH primarily contributes to its antioxidant capacity through the reduction of protein-oxidation adducts [30], whereas total antioxidant capacity (TAC) eliminates non-specific oxidative stressors [31]. Thus, we investigated the serum levels of non-enzymatic components in all participants. When compared to the placebo group, JGT significantly increased total GSH contents and TEAC in the serum (Figure 1C,D, *p* = 0.022 for total GSH contents and *p* = 0.023 for TEAC). Interestingly, total GSH contents in the serum in the JGT group were slightly higher than after 4 weeks of administration (*p* > 0.05), but this was not the case in the placebo group. Similarly, serum TEAC levels in the placebo group did not change during the experiment period, but the levels increased in the JGT group. We also measured enzymatic antioxidant components such as SOD and catalase activities, as shown in Figure 1E,F, and found that the JGT administration increased SOD and catalase activities during the experimental periods in the JGT group when compared to the placebo group (*p* = 0.034 for catalase activities and *p* = 0.047 for SOD activities in between the JGT and placebo groups; *p* = 0.034 for catalase activities and *p* = 0.018 for SOD activities in before vs. after JGT administration in the JGT group). The findings above strongly suggest that JGT works beneficially for CSD patients by restoring and maintaining a balanced redox status, especially by reducing oxidation and preventing antioxidant depletion.

The primary symptoms of dizziness disorders are closely related to the abnormal psycho-emotional status, as seen in phobic, panic, anxiety, depressive, dissociative, or somatoform disorders, as well as the abnormal stress hormone response, which can be found in neuro-otological dysfunctions, dizziness, and vertigo. In this study, when compared to the placebo group, JGT significantly reduced serum cortisol and adrenaline levels in the treatment group (*p* = 0.001 for cortisol and *p* = 0.002 for adrenaline, Figure 2A,B). Serum cortisol levels in the placebo group were slightly elevated, while serum adrenaline levels decreased over the course of the trial. We also looked at serum serotonin and noradrenaline levels because SSRIs and SNRIs are clinically used to treat CSD. When compared to the placebo group, serum serotonin levels increased after JGT treatment in the JGT group (Figure 2D, *p* = 0.046). There was no change in noradrenaline, which is primarily induced by sympathetic nerves, in both groups (Figure 2C).

The HPA axis, a neuroendocrine system that mediates the stress response, regulates the release of stress hormones. Due to HPA axis activation, stress hormones, particularly cortisol, are released in response to stressors to influence various functions in the body [32]. Two catecholamines, adrenaline, and noradrenaline are also intimately linked to this stress response system [33]. The redox system imbalance is closely related to the status of these stress hormones [34]. Unlike these stress hormones, however, serotonin is known to play a soothing role in anxiety, depression, and dizziness [35,36].

Stress is defined as emotionally and physiologically demanding experiences. An appropriate level of stress can be a beneficial stimulus, but excessive stress beyond the human body’s tolerability can lead to pathological consequences [37]. Chronic stress has been shown to activate the HPA axis, raise plasma levels of cortisol, adrenaline, and noradrenaline, and cause oxidative stress [38]. Furthermore, people under chronic stress are more vulnerable to additional stress stimuli, as demonstrated in a study in which women under chronic stress experienced an increase in cortisol levels and elevated serum levels of oxidative stress biomarkers as a result of acute stress [39]. Chronic and excessive stress can also cause and exacerbate neuro-otological disorders by interfering with optimal vestibular compensation [32].

Many patients suffering from chronic dizziness recognize stress as an aggravating factor. However, advising patients to simply avoid stress may exacerbate the situation, either by encouraging excessive passivity or by intensifying anxiety if stress cannot be avoided. Instead, comprehensive information about the causes of symptoms can help patients feel less anxious and better manage their symptoms [40]. More importantly and practically, as there are still unmet healthcare needs in people living with chronic dizziness [20], effective medication to correct the imbalance in the patients’ redox and neuroendocrine systems is required.

Jaeumgeonbi-tang, a Korean medicine herbal mixture recorded in *Donguibogam*, has been widely prescribed to treat a variety of neuro-otological disorders of unknown origin such as headaches, dizziness, and anxiety [41,42]. This study shows that JGT is an effective treatment for CSD patients in terms of reducing oxidative stress, improving antioxidant components, and improving the stress-related hormone system. Meanwhile, herbal mixtures should be used with caution, as unanticipated side effects such as hepatotoxicity or an allergic reaction are possible [43]. In this study, we were able to show that 4 weeks of JGT administration had no side effects or adverse effects, allergic reactions, or hepatotoxicity issues (Table 1 and Table 2).

Although the JGT prescription is complicated as it is composed of 18 herbs, the excellent clinical effects of JGT have been confirmed at Daejeon University Affiliated Korean Medicine Hospitals over the past 20 years. As a result, we conducted this clinical trial to investigate its pharmacological properties. In the future, we will conduct additional research by fractionating the JGT extract to determine which components of the JGT truly exhibit the best pharmacological activity. We hope that our research will aid in the resolution of chronic dizziness management issues and the development of new therapeutic drugs for CSD patients.

## 4. Materials and Methods

### 4.1. Study Design

To investigate the effect of JGT on CSD patients, this study was designed as a randomized, double-blind, parallel-group, placebo-controlled trial. This trial was completed in Dunsan Korean Medicine Hospital of Daejeon University (now: Daejeon Korean Medicine Hospital of Daejeon University) following Institutional Review Board (IRB) approval from the Institutional Review Board of Daejeon Oriental Medical Centre (now: Institutional Review Board of Daejeon Korean Medicine Hospital of Daejeon University, authorization no.: DJOMC-58-Ver.2.00). The current manuscript’s clinical trial registry number can be found in CRIS (Clinical Research Information Service, no.: KCT0000095) and ClinicalTrials.gov’s Protocol Registration and Results (PRS) system (no.: NCT05482828). The trial followed the ethical standards outlined in the Helsinki and Tokyo Declarations.

### 4.2. Participants

Subjects with CSD (aged 20 to 65 years) who were able to fully comprehend the general protocol of this study and voluntarily agreed to participate were included. The detailed inclusion and exclusion criteria are shown in Table 1. The dizziness handicap inventory (DHI) score of 24 points or higher was met by all participants.

To determine whether the participants met any of the exclusion criteria, the medical history, magnetic resonance imaging (MRI), physical examination, and laboratory (hematological and biochemical) tests were used. Patients with inner ear disease (such as benign paroxysmal positional vertigo, vestibular neuritis, Meniere’s disease, etc.), recent stroke (within the last 6 months), hypoglycemia, heart disease, functional dyspepsia, and allergic diseases were excluded from this study. In addition, patients who used medications (such as anticonvulsants, sedatives, antidepressants, sleeping pills, prostate medicine, Parkinson’s drugs, dementia drugs, etc.) that could influence the result of the study were also excluded. Subjects in this study did not have any serious medical problems other than dizziness at the time of study participation, so no special care was required.

After meeting the criteria, all subjects provided written informed consent. A total of 50 participants (45 females, 5 males, mean age 44.9 ± 10.8 years) were recruited and randomly assigned to one of the two groups (JGT or placebo group), using computer-generated block randomization with an allocation ratio of 1:1 and a block size of 4 that had been pre-programmed using SAS Version 9.4 software (SAS Institute., Cary, NC, USA). The mean age of the participants was 45.2 ± 10.7 (23 females, 2 males) in the JGT group and 44.5 ± 11.0 (22 females, 3 males) in the placebo group. In our previous study of 70 dizziness patients [16], there were 16 males and 54 females, indicating a relatively high proportion of women. In this study, dizziness patients who further met the criteria for chronic subjective dizziness were recruited, and we discovered that the proportion of women was higher. According to the findings, women are more likely than men to suffer from chronic subjective dizziness. The detailed baseline characteristics of participants in the JGT and placebo groups are shown in Table 2.

### 4.3. Intervention

The JGT group received JGT tablets, while the placebo group received placebo tablets. Every day, the patient administered a total of 24 g of the daily dose of tablets to himself/herself (three times a day for 4 weeks, a single dose was 8 g). All subjects returned to the clinical trial center once a week after finishing their medications. To prevent drug abuse and track medication adherence, they were asked to return any unused JGT or placebo tablets.

The daily dose of JGT was determined based on our previous research [16]. From 2004 to 2009, 70 dizziness patients who visited Dunsan Korean Medicine Hospital of Daejeon University and did not take other dizziness-related drugs were asked to take JGT decocted with a daily dosage of approximately 79.3 g of medicinal herbs. DHI and visual analog scale (VAS) scores were significantly reduced in the majority of patients. Therefore, we used the same daily dosage of 79.3 g of medical herbs in this study. A pharmaceutical company obtained approximately 21.2 g of dry extract from 79.3 g of medicinal material, and 24 tablets (1 g/tablet), were prepared with a minimum of excipients, and this was determined as the daily intake. The daily dose in this study was 24 g, regardless of participant age. The average treatment period for JGT intake was about 20 days for the 70 patients. Based on this information, the study period was set at four weeks.

In addition, from 2002 to 2010, over 1400 dizziness patients visited Daejeon University Affiliated Korean Medicine Hospitals and were treated with JGT for periods ranging from 5 days to 130 days or longer. JGT was decocted using approximately 80 to 160 g of medicinal herbs as a daily dosage. There were no specific side effects reported by the patients. There have been no reports since of adverse effects, allergic reactions, or liver damage.

#### 4.3.1. Preparation of JGT and Placebo

Drugs were prepared in tablet form by Hanpoong Pharm & Food Co., Ltd. (Daejeon, South Korea). The JGT tablet used in this study was made up of the following eighteen herbs: *Gastrodia elata* Bl. (Gastrodiae Rhizoma), *Adenophora triphylla* var. *japonica* Hara (Adenophorae Radix), *Citrus unshiu* Markovich (Citrus Pericarpium), *Atractylodes japonica* Koidz. (Atractylodis Rhizoma Alba), *Pinellia ternata* (Thunb.) Breitenbach (Pinelliae Rhizoma), *Poria cocos* Wolf (Hoelen), *Paeonia lactiflora* Pall. (Paeoniae Radix Alba), *Rehmannia glutinosa* Liboschitz (Rehmanniae Radix), *Angelica gigas* Nakai (Angelicae Gigantis Radix), *Liriope platyphylla* Wang et Tang (Liriopis Tuber), *Cnidium officinale* Makino (Cnidii Rhizoma), *Zizyphus jujuba* Miller var. *inermis* Rehder (Zizyphi Fructus), *Zingiber officinale* Rosc. (Zingiberis Rhizoma), *Schizonepeta tenuifolia* Briquet (Schizonepetae Spica), *Saposhnikovia divaricata* Schischkin (Saposhnikoviae Radix), *Polygala tenuifolia* Willd. (Polygalae Radix), *Glycyrrhiza uralensis* Fisch. (Glycyrrhiza Radix), and *Mentha arvensis* Linne var. *piperascens* Malinvaud (Menthe Herba) (Appendix A). The placebo tablet was manufactured at the same dosage size as JGT and was made up of the following ingredients: 50% corn starch, 49.45% lactose, 0.5% caramel color, and 0.05% Ssanghwa-tang scent.

Based on the daily dosage of 79.3 g, 11.56 g of Gastrodiae Rhizoma, 11.56 g of Ade-nophorae Radix, 5.58 g of Hoelen, 3.72 g of Citrus Pericarpium, 3.72 g of Pinelliae Rhi-zoma, 3.72 g of Paeoniae Radix Alba, 3.72 g of Rehmanniae Radix, 3.72 g of Angelicae Gigantis Radix, 3.72 g of Liriopis Tuber, 3.72 g of Cnidii Rhizoma, 3.72 g of Zizyphi Fructus, 3.72 g of Zingiberis Rhizoma, 3.72 g of Schizonepetae Spica, 3.72 g of Saposhnikoviae Radix, 1.86 g of Polygalae Radix, 1.86 g of Glycyrrhiza Radix, 0.21 g of Menthe Herba, and 5.78 g of Atractylodis Rhizoma Alba were placed into the extraction tank. Purified water, 10 times the amount of the herbs, was then added. After 3 h of extraction at 80–100 °C, the extract was filtered through 50 mesh. The filtrate was concentrated under reduced pressure at 60 °C or less to produce dry extract (yield approximately 26.04%), and an excipient was added before coating the dried extract to produce a tablet form.

#### 4.3.2. Fingerprinting Analysis of JGT

For quantitative analysis, a solution containing seven different types of reference compounds (albiflorin, paeoniflorin, nodakenin, hesperidin, liquirtigenin, glycyrrhizic acid, and decursin) was prepared in 90% methanol at a concentration of 200 μg/mL and stored at <4 °C. The standard solutions were prepared at six different concentrations using a dilution method (methanol). The peak areas at six concentrations in the range of 7.8–250 μg/mL for all reference compounds were measured to attain the calibration curves. To calculate the contents of the main JGT components, the linearity of the peak area (*y*) versus concentration (*x*, μg/mL) for each component was used.

Under identical conditions, quantitative analysis was carried out using an 1100 series HPLC device (Agilent Technologies, Santa Clara, CA, USA) equipped with an autosampler (G11313A), column oven (GA1316A), binary pump (G1312), diode-array detector, and degasser (GA1379A). During the analysis, the analytical column with a Gemini C18 (4.6 × 250 mm; particle size, 5 μm; Phenomenex, Torrance, CA, USA) was kept at 30 °C. ChemStation software (Agilent Technologies) was used to collect and process data. The mobile phase conditions included 10% acetonitrile in water with (A) 0.05% formic acid and (B) 0.1% acetonitrile in water. The gradient flow was as follows: 0–30 min, 0–20% B; 30–50 min, 20–75% B; and 50–60 min, 75–100% B. The analysis was carried out at a flow rate of 1.0 mL/min and detected at wavelengths ranging from 230 to 330 nm. The injection volume was 10 μL.

A total of 7 kinds of chemical constructions were identified from JGT under various UV wavelengths (230, 254, 280, and 330 nm). Albiflorin and paeoniflorin were detected under 230 nm and their retention times were 14.3 and 15.8 min, respectively. Nodakenin and decursin were detected at 330 nm with 23.7 min and 50.4 min of retention time, respectively. Furthermore, hesperidin and liquiritigenin were observed at 280 nm, with retention times of 26.9 and 33.4 min, respectively. Only glycyrrhizinic acid was detected with a retention time of 45.3 min at 254 nm. Paeoniflorin was the most abundant chemical compound in JGT (92.4079 ± 0.3724 μg/mg, Figure 3 and Appendix A).

### 4.4. Outcome Measurement

The intervention assignments were undisclosed to the outcome investigators and statistical analysts until the trial’s conclusion. For pathophysiological parameters, fasting blood samples were collected from each subject at the start and end of the trial. The process of this trial is summarized in Figure 4.

A trained phlebotomist collected whole blood samples from all participants and placed them in Vacutainer^®^ tubes. Blood samples collected in EDTA-containing tubes were incubated at room temperature (RT) for fibrinogen removal before being centrifuged at 3000× *g* for 2 min at 4 °C. Then, the whole serum samples were transferred to microtubes for multiple aliquots before being stored at −70 °C until analysis. NO, MDA (measured by TBARS assay), total GSH contents, TEAC, catalase, and SOD activities were all measured in the serum samples.

Serum NO levels were analyzed using a commercial kit and the manufacturer’s instructions (Nitric Oxide Colorimetric Assay Kit, BioVision, Milpitas, CA, USA, Catalog no.: K262). Serum levels of MDA, a byproduct of oxidative stress, were measured by TBARS assay using the previous method [44]. In brief, 75 μL of serum sample was added to microcentrifuge tubes and mixed with 3 μL of butylated hydroxytoluene in methanol. Next, 75 μL of 1M phosphoric acid and an equal volume of 2-thiobarbituric acid were added and then mixed in the same tube using the vortex. The tubes were then incubated for 60 min at 60 °C. Following incubation, the mixture was added to a microplate (100 μL of each), and the absorbance at 535 and 572 nm (to correct for baseline absorption) was measured with a spectrophotometer (Versa Max. Molecular Device, Sunnyvale, CA, USA). Tetra-methoxy-propane was used as a standard solution for the quantification analysis calibration curve.

The GSH levels in the serum were determined using a previously described method [45]. Briefly, in a 96-well plate, either 50 μL of dilute serum (in 10 mM PBS, pH 7.2) or reduced GSH (as a standard) was mixed with 80 μL DTNB/NADPH mixture (10 μL, 4 mM DTNB, and 70 μL, 0.3 mM NADPH). Then, to each well in the plate, 20 μL (0.06 U) of GSH-reductase solution was added, and absorbance at 405 nm was measured using a plate reader. The serum antioxidant capacity was determined using the TEAC method which was slightly modified from the previous description [46]. Amounts of 90 μL of 10 mM PBS (pH 7.2), 50 μL of myoglobin solution (18 μM), 20 μL of 3 mM ABTS solution, 20 μL of a dilute serum sample, or different concentrations of Trolox (as a standard) were mixed together for 3 min at 25 °C in a 96-well microplate. After then, each well received 20 °C H_2_O_2_ and was incubated for 5 min. At 600 nm, the absorbance was measured with a plate reader.

Serum SOD activity was measured with a commercial SOD assay kit (Dojindo Laboratories, Kumamoto, Japan) and the manufacturer’s instructions. Bovine erythrocyte SOD (Sigma, St. Louis, MO, USA) was used to construct a calibration curve (0 to 50 U/mL). Serum catalase activities were measured as previously described [47]. For catalase activity analysis, a total of 33 μL mixed working buffer (i.e., a mixture of PHS (15 μL, 250 mM, pH 7.2), methanol (15 μL, 12 mM), and H2O2 (3 μL, 44 mM)), was added to 96-well plates. Then, 30 μL of each sample/standard solution was added, and the reaction was carried out for 10–20 min at RT. The reaction was halted by the addition of 45 μL Purpald solution (22.8 mM Purpald in 2 N potassium hydroxide). The samples were then incubated at RT for 20 min before adding 15 μL potassium periodate (65.2 mM in 0.5 N potassium hydrate). Absorbance at 550 nm was measured using a spectrophotometer.

Cortisol, adrenaline, noradrenaline, and serotonin levels in serum were measured using ELISA kits (cortisol, adrenaline, and serotonin, LDN GmbH & Co., KG, Nordhorn, Germany) and the manufacturer’s protocol. A spectrophotometer was used to measure absorbance.

### 4.5. Statistical Analysis

Statistical comparisons of values among groups and changes in values between groups were carried out with Student’s *t*-test utilizing the PASW Statistics 20 for Windows (SPSS Inc., Chicago, IL, USA). Levels of statistical significance were reported at *p* < 0.05 and *p* < 0.01.

## 5. Conclusions

CSD is a neuro-otological disorder characterized by persistent non-vertiginous dizziness and unsteadiness lasting more than three months and exacerbated by upright posture, complex visual stimuli, or demanding visual tasks. To date, SSRI or SNRI has been used for treatment, but it is frequently discontinued due to side effects and the patient’s high drug sensitivity, even apparent at lower doses than those used for depression. Because there is a need for a new effective and safe drug, we investigated how JGT works within the mechanism of CSD in this trial. Prior to the trial, seven different types of chemicals were identified from JGT using quantitative analysis. Fifty CSD patients were randomly allocated to the JGT or placebo groups and were instructed to take either JGT or placebo for four weeks, respectively. JGT’s effects on oxidative stress biomarkers, endogenous antioxidants, stress-related hormones, and drug safety were studied using blood tests before and after administration.

In terms of oxidative stress biomarkers, NO levels were reduced similarly in both groups, but TBARS levels, a final oxidation product, were significantly lower after JGT administration in the JGT group compared to the placebo group. When compared to the placebo group, the endogenous antioxidant components GSH, TEAC, catalase, and SOD were significantly improved in the JGT group after JGT administration. Cortisol and adrenaline levels were significantly lower after treatment in the JGT group compared to the placebo group, but there was no difference in noradrenaline levels. Serotonin levels were significantly elevated in the JGT group compared to the placebo group after treatment. In terms of drug safety, biochemical and hematological tests were performed to assess hepatotoxicity, and no abnormalities were discovered.

Thus, the study found that JGT has a beneficial effect on the redox system and stress hormone indicators in CSD patients. More research into how JGT functions in the pathophysiological mechanisms of CSD is required.

## Figures and Tables

**Figure 1 pharmaceuticals-15-01375-f001:**
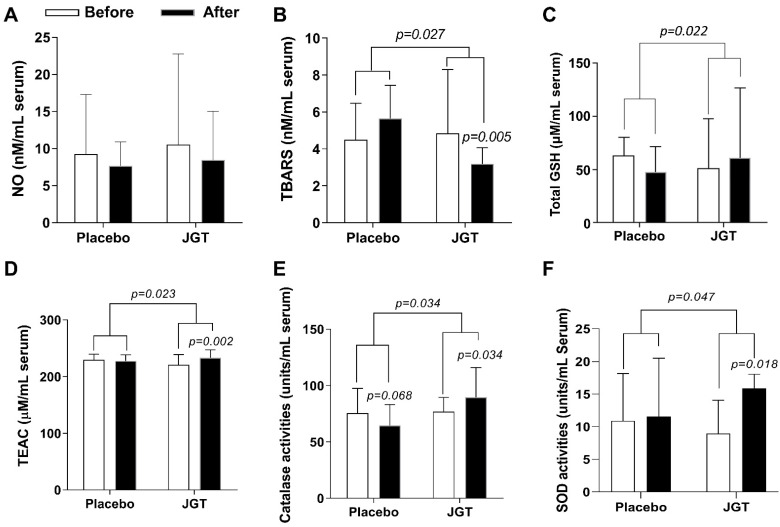
Effects of JGT on the oxidative stress and antioxidant components in the serum levels. Serum levels of (**A**) NO (nitric oxide), (**B**) MDA (malondialdehyde, measured by thiobarbituric acid reactive substances (TBARS) assay), (**C**) total GSH (glutathione), (**D**) TEAC (Trolox equivalent antioxidant capacity), (**E**) catalase activities, and (**F**) SOD (superoxide dismutase) activities were measured after the completion of trials. Data were expressed as mean ± SD. *p*-values were calculated by comparing before and after treatment changes within each group and between the two groups.

**Figure 2 pharmaceuticals-15-01375-f002:**
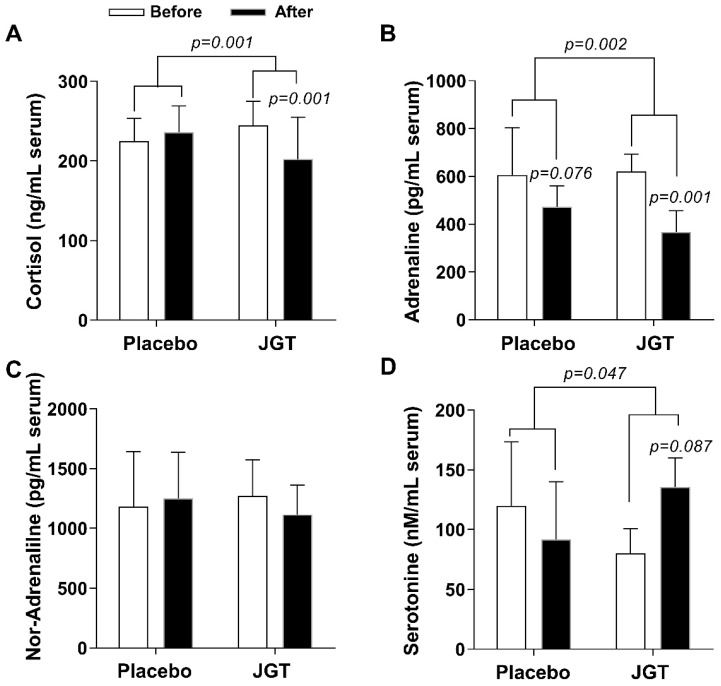
Effects of JGT on stress-related hormone release in serum levels. Serum levels of (**A**) cortisol, (**B**) adrenaline, (**C**) noradrenaline, and (**D**) serotonin were measured after the completion of trials. Data were expressed as mean ± SD. The *p*-values were calculated by comparing before and after treatment changes within each group and between the two groups.

**Figure 3 pharmaceuticals-15-01375-f003:**
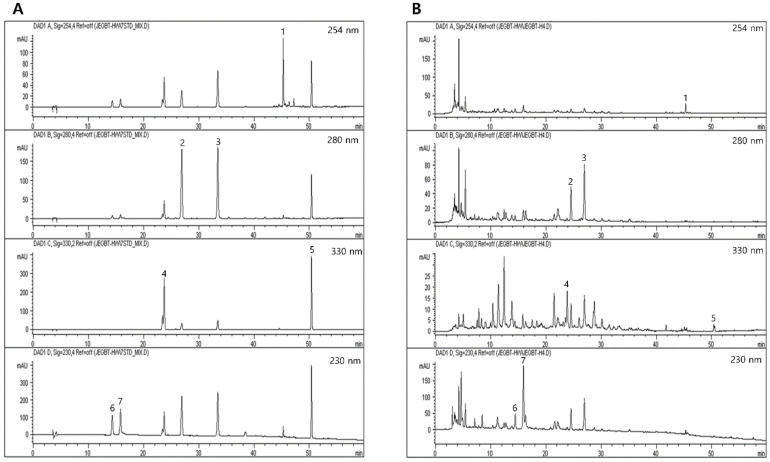
HPLC chromatogram of JGT for fingerprinting analysis. JGT and its reference standards chemical compounds were subjected to HPLC analysis. Fingerprinting analysis of (**A**) a total of 7 kinds of reference chemical compounds and (**B**) JGT. 1. Glycyrrhizinic acid; 2. hesperidin; 3. liquritigenin; 4. nodakenin; 5. decursin; 6. albiflorin; and 7. paeoniflorin, respectively.

**Figure 4 pharmaceuticals-15-01375-f004:**
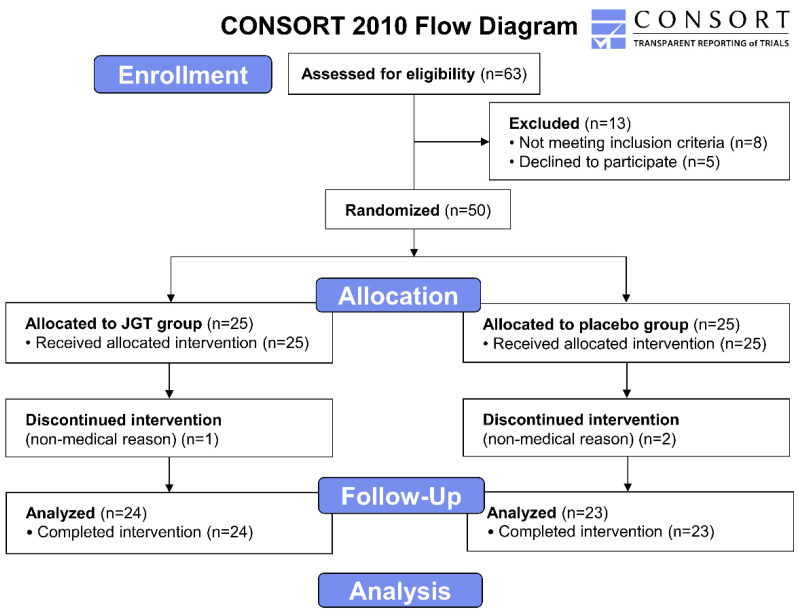
CONSORT 2010 flow diagram of the clinical trial. The phases of the trial, screening, enrolment, randomization, follow up, and data analysis are demonstrated on the flow diagram. A total of 63 subjects diagnosed with CSD were recruited at first. After the exclusion of 13 subjects, 50 subjects in total were divided between the JGT group (*n* = 25) and placebo group (*n* = 25), and JGT or placebo was administrated to all participants in each group for 4 weeks. The assay results of the participants who completed the procedure (*n* = 24 for the placebo group and *n* = 23 for the JGT group) were included to be analyzed. CSD: chronic subjective dizziness; JGT: *Jaeumgeonbi-tang*.

**Table 1 pharmaceuticals-15-01375-t001:** Inclusion and exclusion criteria.

Inclusion Criteria	Exclusion Criteria
CSD (Chronic subjective dizziness) patients, aged from 20 to 65 DHI (Dizziness handicap inventory) scores ≥ 24 Those who can fully comprehend the general protocol of this study and voluntarily agree to participate	Inner ear disease (benign paroxysmal positional vertigo, vestibular neuritis, Meniere’s disease, etc.) Dizziness secondary to specific diseases such as hypoglycemia, recent stroke (within the last 6 months), or heart disease. Use of medications that could influence the result of the study (anticonvulsants, sedatives, antidepressants, sleeping pills, prostate medicine, Parkinson’s drugs, dementia drugs, etc.) Pregnancy, breastfeeding, or plans of becoming pregnant Functional dyspepsia (persistent, recurring abdominal pain, or discomfort) Other allergic diseases Ineligibility for other reasons in the opinion of the research clinician (when the physician determines that there are significant physical or mental defects that the patient cannot understand and follow the protocol)

**Table 2 pharmaceuticals-15-01375-t002:** Baseline characteristics of participants.

Contents	JGT ^1^	Placebo	Total
Number and sex ratio (M:F)	25 (2:23)	25 (3:22)	50 (5:45)
Age (years)	30~60 (45.2 ± 10.7)	21~61 (44.5 ± 11.0)	21~61 (44.9 ± 10.8)
Height (cm)	160.6 ± 6.1	160.1 ± 5.9	160.5 ± 5.9
Body weight (kg)	60.7 ± 10.0	56.7 ± 8.8	58.6 ± 9.5
BMI ^2^ (kg/m^2^)	23.6 ± 3.1	22.0 ± 2.8	22.7 ± 3.0
Pulse (rate/min)	71.1 ± 8.8	70.2 ± 10.1	71.2 ± 9.4
SBP ^3^ (mmHg)	112.9 ± 12.9	114.1 ± 13.5	113.5 ± 14.6
DBP ^4^ (mmHg)	73.9 ± 12.5	76.7 ± 12.5	75.3 ± 11.9

^1^ JGT: *Jaeumgeonbi-Tang*; ^2^ BMI: body mass index; ^3^ SBP: systolic blood pressure; ^4^ DBP: diastolic blood pressure. A total of 50 participants were enrolled in this randomized, double-blind, controlled clinical trial for 4 weeks. After 4 weeks of intervention, a total of 47 participants (*n* = 24 for the control and *n* = 23 for the JGT group, respectively) had completed the trial and were included in the analysis. Before the initiation of the intervention, we investigated the baseline characteristics of the participants. Data are expressed as Mean ± SD.

## Data Availability

Data is contained within the article and Appendix A.

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
