# Peer review of "Pharmacological Properties of Jaeumgeonbi-Tang on Redox System and Stress-Related Hormones in Chronic Subjective Dizziness: A Randomized, Double-Blind, Parallel-Group, Placebo-Controlled Trial"

_pharmaceuticals, 2022, doi:10.3390/ph15111375_

Round 1

Reviewer 1 Report

This article “Pharmacological Properties of Jaeumgeonbi-Tang on Redox Sys-2 tem and Stress-related Hormones in Chronic Subjective Dizzi-3 ness: A Randomized, Double-blind, Parallel-group, Placebo-4 controlled Trial” by Chan-young Kim and co-author described Jaeumgeonbi-Tang (JGT) as traditional herbal medicine and its randomized, double-blind, parallel-group and placebo-controlled trial with a total of 50 CSD patients. The data presented here has some potential value for future therapeutic validation. However, a lot of information is missing in the present form. Please address the comets.

(1) Where is the dosimetry data? How do you confirm the 24 g daily dose is the optimal dose?

(2) What are the off-target/side effects at this dosing level?

(3) Is there any special care needed during the treatment study?

(4) Is there any co-relation between the daily dose and patient age?

(5) The treatment study went on for 4 weeks. How you optimized this time point? What about the other time points?

(6) The JGT Drugs were prepared in tablet form by Hanpoong Pharm & Food Co., Ltd. The JGT tablet used in this study has 18 herbs; please detail the percentage of each ingredient and describe the preparation procedure with standard qualifying criteria for each herb.

 (7) Is there any particular reason for the gender ratio (45 females vs. 5 males)?

Author Response

This article “Pharmacological Properties of Jaeumgeonbi-Tang on Redox System and Stress-related Hormones in Chronic Subjective Dizziness: A Randomized, Double-blind, Parallel-group, Placebo-controlled Trial” by Chan-young Kim and co-author described Jaeumgeonbi-Tang (JGT) as traditional herbal medicine and its randomized, double-blind, parallel-group and placebo-controlled trial with a total of 50 CSD patients. The data presented here has some potential value for future therapeutic validation. However, a lot of information is missing in the present form. Please address the comets.

-> Thank you for thoroughly reviewing the article and providing us with helpful comments. We have modified the article thoroughly based on the comments.

(1) Where is the dosimetry data? How do you confirm the 24 g daily dose is the optimal dose?

-> The daily dose of JGT was determined based on our previous research ([45] Lee, J.H.; Shin, H.S.; Kim, D.H.; Jo, C.H.; Lim, S.M.; An, J.J.; Jo, H.K.; Kim, Y.S.; Seol, I.C.; Yoo, H.R. Statistical Study in 70 Cases for Dizziness Patients on the Effect of Jaeumgeonbi-tang Gamibang. J. Physiol. & Pathol. Korean Med. 2010, 24, 171-176.). From 2004 to 2009, 70 dizziness patients who visited Dunsan Korean Medicine Hospital of Daejeon University and did not take other dizziness-related drugs were asked to take JGT decocted with a daily dosage of approximately 79.3 g of medicinal herbs. DHI and visual analog scale (VAS) scores were significantly reduced in the majority of patients. Therefore, we used the same daily dosage of 79.3 g of medical herbs in this study. A pharmaceutical company obtained approximately 21.2 g of dry extract from 79.3 g of medicinal material, and 24 tablets (1 g/tablet), were prepared with a minimum of excipients, and this was determined as the daily intake. (added -- page: 9/15, lines: 343-350)

(2) What are the off-target/side effects at this dosing level?

-> From 2002 to 2010, over 1,400 dizziness patients visited Daejeon University Affiliated Korean Medicine Hospitals and were treated with JGT for periods ranging from 5 days to 130 days or longer. JGT was decocted using approximately 80 to 160 g of medicinal herbs as a daily dosage. There were no specific side effects reported by the patients. There have been no reports of adverse effects, allergic reactions, or liver damage. (added -- page: 9/15, lines: 354-358)

(3) Is there any special care needed during the treatment study?

-> Subjects in this study did not have any serious medical problems other than dizziness at the time of study participation, so no special care was required. (added -- page: 8/15, lines: 312-313)

(4) Is there any co-relation between the daily dose and patient age?

-> The daily dose in this study was 24 g, regardless of participant age. (added -- page: 9/15, lines: 351-352)

(5) The treatment study went on for 4 weeks. How you optimized this time point? What about the other time points?

-> The average treatment period for JGT intake was about 20 days for the 70 patients. Based on this information, the study period was set at four weeks. (added -- page: 9/15, lines: 352-353)

(6) The JGT Drugs were prepared in tablet form by Hanpoong Pharm & Food Co., Ltd. The JGT tablet used in this study has 18 herbs; please detail the percentage of each ingredient and describe the preparation procedure with standard qualifying criteria for each herb.

-> Based on the daily dosage of 79.3 g, 11.56 g of Gastrodiae Rhizoma, 11.56 g of Adenophorae Radix, 5.58 g of Hoelen, 3.72 g of Citrus Pericarpium, 3.72 g of Pinelliae Rhizoma, 3.72 g of Paeoniae Radix Alba, 3.72 g of Rehmanniae Radix, 3.72 g of Angelicae Gigantis Radix, 3.72 g of Liriopis Tuber, 3.72 g of Cnidii Rhizoma, 3.72 g of Zizyphi Fructus, 3.72 g of Zingiberis Rhizoma, 3.72 g of Schizonepetae Spica, 3.72 g of Saposhnikoviae Radix, 1.86 g of Polygalae Radix, 1.86 g of Glycyrrhiza Radix, 0.21 g of Menthe Herba, and 5.78 g of Atractylodis Rhizoma Alba were placed into the extraction tank. Purified water, 10 times the amount of the herbs, was then added. After 3 hours of extraction at 80~100 °C, the extract was filtered through 50 mesh. The filtrate was concentrated under reduced pressure at 60 °C or less to produce dry extract (yield approximately 26.04%), and an excipient was added before coating the dried extract to produce a tablet form. (added -- page: 9/15, lines: 375-385)

(7) Is there any particular reason for the gender ratio (45 females vs. 5 males)?

-> In our previous study of 70 dizziness patients research ([45] Lee, J.H.; Shin, H.S.; Kim, D.H.; Jo, C.H.; Lim, S.M.; An, J.J.; Jo, H.K.; Kim, Y.S.; Seol, I.C.; Yoo, H.R. Statistical Study in 70 Cases for Dizziness Patients on the Effect of Jaeumgeonbi-tang Gamibang. J. Physiol. & Pathol. Korean Med. 2010, 24, 171-176.), there were 16 males and 54 females, indicating a relatively high proportion of women. In this study, dizziness patients who further met the criteria for chronic subjective dizziness were recruited, and we discovered that the proportion of women was higher. According to the findings, women are more likely than men to suffer from chronic subjective dizziness. (added -- page: 8/15, lines: 321-326)

Reviewer 2 Report

Kim et al. studied the pharmacological effects of traditional herbal medicine called as Jaeumgeonbi-Tang (JGT) in a randomized, double-blind, parallel group, placebo-controlled trial. It is an interesting study. The authors planned this study very well however, there are some comments that authors should justify or include in the revised manuscript prior to its acceptance.

1.      Please justify the variation in the numbers of males and females (5:45).

2.      How does block randomization occur? Write the name of the program.

3.      How was the dose decided (24 g daily for all age groups)? Authors should write the differences in doses in reference to the patient’s age.

4.      Authors should check the effects of JGT doses on liver and kidney parameters in reference to the ages of participants.

5.      As JGT is composed of several herbs. I understand, it would be very difficult to know, which herb is really showing the best pharmacological activity for CSD but, if possible, authors should analyze such composition in more detail as it will reduce the amount of dose and will be easily taken by patients.

6.      Overall, this study is very well written. The experiments were carried out properly and the authors represented the data significantly. 

Author Response

Kim et al. studied the pharmacological effects of traditional herbal medicine called as Jaeumgeonbi-Tang (JGT) in a randomized, double-blind, parallel-group, placebo-controlled trial. It is an interesting study. The authors planned this study very well however, there are some comments that authors should justify or include in the revised manuscript prior to its acceptance.

-> Thank you for thoroughly reviewing the article and providing us with helpful comments. We have modified the article thoroughly based on the comments.

  1. Please justify the variation in the numbers of males and females (5:45).

-> In our previous study of 70 dizziness patients ([45] Lee, J.H.; Shin, H.S.; Kim, D.H.; Jo, C.H.; Lim, S.M.; An, J.J.; Jo, H.K.; Kim, Y.S.; Seol, I.C.; Yoo, H.R. Statistical Study in 70 Cases for Dizziness Patients on the Effect of Jaeumgeonbi-tang Gamibang. J. Physiol. & Pathol. Korean Med. 2010, 24, 171-176.), there were 16 males and 54 females, indicating a relatively high proportion of women. In this study, dizziness patients who further met the criteria for chronic subjective dizziness were recruited, and we discovered that the proportion of women was higher. According to the findings, women are more likely than men to suffer from chronic subjective dizziness. (added -- page: 8/15, lines: 321-326)

  1. How does block randomization occur? Write the name of the program.

-> A total of 50 participants (45 females, 5 males, mean age 44.9 ± 10.8 years) were recruited and randomly assigned to one of the two groups (JGT or placebo group), using computer-generated block randomization with an allocation ratio of 1:1 and a block size of 4 that had been pre-programmed using SAS Version 9.2 software (SAS Instute, Inc., Cary, NC, USA). (added -- page: 8/15, lines: 318-319)

  1. How was the dose decided (24 g daily for all age groups)? Authors should write the differences in doses in reference to the patient’s age.

-> The daily dose of JGT was determined based on our previous research ([45] Lee, J.H.; Shin, H.S.; Kim, D.H.; Jo, C.H.; Lim, S.M.; An, J.J.; Jo, H.K.; Kim, Y.S.; Seol, I.C.; Yoo, H.R. Statistical Study in 70 Cases for Dizziness Patients on the Effect of Jaeumgeonbi-tang Gamibang. J. Physiol. & Pathol. Korean Med. 2010, 24, 171-176.). From 2004 to 2009, 70 dizziness patients who visited Dunsan Korean Medicine Hospital of Daejeon University and did not take other dizziness-related drugs were asked to take JGT decocted with a daily dosage of approximately 79.3 g of medicinal herbs. DHI and visual analog scale (VAS) scores were significantly reduced in the majority of patients. Therefore, we used the same daily dosage of 79.3 g of medical herbs in this study. A pharmaceutical company obtained approximately 21.2 g of dry extract from 79.3 g of medicinal material, and 24 tablets (1 g/tablet), were prepared with a minimum of excipients, and this was determined as the daily intake. The daily dose in this study was 24 g, regardless of participant age. (added -- page: 9/15, lines: 343-352)

  1. Authors should check the effects of JGT doses on liver and kidney parameters in reference to the ages of participants.

-> We further checked the effect of JGT on liver and kidney parameters in reference to the ages of the subjects. Because the subjects’ ages ranged from 20 to 62, we divided the subjects into two groups: those aged 20 to 45 and those aged 46 to 62. We used Wilcoxon signed rank test to determine whether liver and kidney parameters in each group differed before and after taking JGT. After 4 weeks of intervention in the JGT group, the increase in total protein in the blood was statistically significant in the 20-45 group, but the change was within the normal range. After 4 weeks of intervention in the placebo group, the increase in total protein in the blood and the decrease in A/G ratio were statistically significant in the 46-62 group, but they were also within the normal range. Other items revealed no statistically significant results. The above differences may have been statistically significant because of the results of multiple tests. (added – page: 5/15, lines: 160-170 & * We’ve also added details in the Supplementary Table S1)

  1. As JGT is composed of several herbs. I understand, it would be very difficult to know, which herb is really showing the best pharmacological activity for CSD but, if possible, authors should analyze such composition in more detail as it will reduce the amount of dose and will be easily taken by patients.

-> Although the JGT prescription is complicated, the excellent clinical effects of JGT have been confirmed at Daejeon University Affiliated Korean Medicine Hospitals over the past 20 years. As a result, we conducted this clinical trial to investigate its pharmacological properties. In the future, we will conduct additional research by fractionating the JGT extract to determine which components of the JGT truly exhibit the best pharmacological activity. (added – page: 7/15, lines: 280-285)

  1. Overall, this study is very well written. The experiments were carried out properly and the authors represented the data significantly.

-> Thank you.

Round 2

Reviewer 1 Report

Accept in present form